# Double-Duty Carers’ Health and Wellbeing during COVID-19: Exploring the Role of Mobility of the Care Economy in Southern Ontario, Canada

**DOI:** 10.3390/ijerph21060730

**Published:** 2024-06-04

**Authors:** Léa Ravensbergen, Sakshi Mehta, Bharati Sethi, Catherine Ward-Griffin, Allison Williams

**Affiliations:** 1School of Earth, Environment & Society, McMaster University, Hamilton, ON L8S 4L8, Canada; awill@mcmaster.ca; 2Faculty of Health Sciences, McMaster University, Hamilton, ON L8S 4L8, Canada; sakshimt@gmail.com; 3The Department of Political Studies, Trent University Peterborough, Peterborough, ON K9L 0G2, Canada; bharatisethi@trentu.ca; 4Faculty of Health Sciences, Western University, London, ON N6A 3K7, Canada; cwg@uwo.ca

**Keywords:** mobility of the care economy, double-duty carers, carer–employee, COVID-19, health, wellbeing, interviews

## Abstract

Double-Duty Carers (DDCs) refer to people who work in the healthcare industry while also providing unpaid care to relatives, friends, or neighbours. The expectations placed on DDCs is expected to grow, and these employees already experience a high caring burden. As such it is important to understand how best to support their health and wellbeing. This paper explores DDCs’ wellbeing during the COVID-19 pandemic, focusing an understudied factor: their mobility constraints. Following the Mobility of the Care Economy framework and a qualitative research design, it does so through a thematic analysis of 16 semi-structured interviews with female DDCs in Southern Ontario, Canada. Once data saturation was reached, three mobility pathways during the pandemic were identified, all of which negatively affected DDCs wellbeing. First, some COVID-19 policies (e.g., testing requirements) resulted in increased mobility demands and increased spatiotemporal constraints. Second, the closure of institutions that care for dependents (schools, daycares, day centres) resulted in forced reduced mobility, which increased financial stress. Finally, indirect mobility effects were identified: the reduced mobility of other informal carers increased the workload and emotional strain on DDCs. The paper concludes with a discussion of mobility-related policies that could improve DDC wellbeing.

## 1. Introduction

In the healthcare sector, many workers are believed to be “double-duty carers” (DDCs) [1,2,3]. DDCs provide paid care through their employment in healthcare, providing frontline services, homecare, long-term care, or institutional care. What makes DDCs unique compared to other healthcare employees is that they also provide unpaid care to dependants outside of work, including relatives, friends, or neighbours. The people they provide unpaid care for are often sections of the population that are more likely to need care, such as children, older adults, or people with disabilities. As such, they are a subset of carer–employees, those who work in any kind of paid employment while also providing unpaid care.

Traditionally, caring is gendered labour, with most formal and unpaid carers continuing to be female. For example, in Canada, approximately 80% of healthcare workers are female [4], while the majority of unpaid caring roles are traditionally filled by women [1]. Familial expectations, obligation because of professional expertise, cultural traditions, and feelings of kinship are factors that disproportionately influence women to adopt unpaid caring roles [5,6].

Unpaid caring work provides an annual contribution of CAD 25 billion to the Canadian economy [7]. Globally, this unpaid care serves as the backbone of the healthcare system [8]. Furthermore, with the growth of home and community care, there will continue to be an increased expectation towards family members or friends to provide unpaid care [9], and given that the DDC role is often adopted due to the carer having professional healthcare expertise [6], it is assumed that the number of DDCs will increase. Due to the high caring burden experienced by DDCs, it is necessary to investigate how their mental and physical health and wellbeing can be best supported.

The COVID-19 pandemic complicated DDCs’ health and wellbeing [10]. Research illustrates that, during the pandemic, 63% of carers reported an increase in unpaid caregiving duties due to a myriad factors, including an increase in the needs of dependents and difficulty accessing health services [11]. Further, 56% reported providing care during the pandemic as more emotionally difficult [11]. Recent studies investigating COVID-19’s impact on caregiving suggest that improving DDCs’ lives and wellbeing will involve targeting policy discussions and increasing support, such as embracing the adoption of carer-friendly workplaces for these populations [12]. This will be vital in supporting front-line DDCs and reducing negative consequences like illness, work absences, stress, and being a source of contagion to those they are caring for.

This study focused on one aspect of DDCs’ unique situation that has received little attention so far: their mobility. Ample research has examined the relationship between travel behaviours and health and wellbeing; in fact, several literature reviews have been written on the topic [13,14,15,16,17]. For instance, a review of travel and subjective wellbeing found that travel behaviour affects wellbeing in five ways: through experiences during travel, through the participation in activities enabled by travel, through the activities during travel, through trips where travel is the activity, and through potential travel [13]. Different types of travel may have different impacts on wellbeing (e.g., commuting or travelling to leisure destinations). Another review examined the relationship between subjective wellbeing and one type of travel, commuting, noting that commutes differ from other forms of travel in their regularity, compulsory nature, and their large time investment [18]. In this review, it was found that research has examined the affective experiences of commuting during travel, satisfaction with the commute, the spill-over effects from the commute to other life domains, and commuting’s impacts on physical health [18].

Commuting, or travel to complete paid labour, is only one type of essential trip. Ample time is also devoted to travelling to complete unpaid labour, which includes other essential trips such as traveling to get groceries [19,20,21,22], to access healthcare [23,24], or to bring children to daycare or school [25,26,27,28]. However, this travel to provide unpaid care has been systematically underquantified and undervalued in transport research. Because women tend to complete the bulk of this travel, many have argued that this oversight is due to a gender bias in how we measure transportation [29,30,31]. To counter this bias, Sánchez de Madariaga [29] coined the term ‘mobility of care’ to refer to all travel required for household upkeep, including grocery shopping, escorting children or older adult dependents, and attending health-related destinations [29]. Although far less studied than mobility for paid employment, recent work quantifying these trips has found that mobility of care represents a far greater proportion of daily travel than anticipated: in many cases, approximately 30% of daily trips are for care purposes [29,30,32].

One study to date has directly examined the relationship between mobility of care and wellbeing. Here, this type of travel was associated with many negative wellbeing outcomes, but only for women [33]. Focusing on working mothers, another recent paper explored mobility of care and wellbeing indirectly by highlighting the ample physical, emotional, and logistical labour required for mobility of care [34]. Given that these two studies did not focus on DCCs, a population known to experience stressful time constraints, it is reasonable to expect that this type of travel will negatively impact DCCs’ subjective wellbeing.

More research on mobility of care, an understudied and significant portion of daily mobility, is needed. Williams et al. [35], however, argue that, for many people, mobility for paid labour and that for unpaid labour cannot be studied separately as they shape one another. They advocate for future research to consider what they term the Mobility of the Care Economy framework (Figure 1). This framework bridges two concepts: the conceptual model of double-duty caregiving (DDC) [5] and the care economy [36,37]. The DDC conceptual model highlights the dynamic process whereby healthcare workers negotiate professional and familial care roles in their lives through the ‘caregiving interface’—an interface which is shaped through two processes: the professionalization of familial care and striving for balance [5]. The care economy, on the other hand, captures care wholistically and broadly. Rather than focusing on paid or unpaid care, it considers both types of labour, as well as the roles of both those providing and those receiving care [37]. Thinking about these two concepts, the mobility of the care economy framework considers all the mobility aspects of this wholistic framing of care: all travel required for both paid and unpaid care amongst both caregivers and those receiving care. Drawing on the original DDC conceptual model, it highlights how resources, expectations, and spatiotemporal constraints/ tensions influence the Mobility of the Care Economy interface.

This paper explores the mobility constraints, some originating from paid labour needs and some from unpaid labour needs, of DDCs during the COVID-19 pandemic and their impact on DCCs’ health and wellbeing. In doing so, this paper aims to answer the following research question: what role does mobility play in shaping DDCs’ health and wellbeing? Although many aspects of the COVID-19 pandemic no longer impact our daily lives, focusing on this period of time can still offer significant insights. First, were another pandemic to occur in the future, or a different catastrophe with similar impacts, results from this work could inform how best to manage future events. Further, the focus on mobility and its impacts on wellbeing in the context of DCCs in unique and likely important to consider when crafting carer-friendly policies during non-pandemic times. Drawing on semi-structured interviews with DDCs in Southern Ontario, Canada, three mobility pathways are identified that negatively impacted DDCs’ wellbeing: increased mobility demands, forced decreased mobility, and the indirect impacts of mobility.

## 2. Materials and Methods

### 2.1. Methods

This mixed-methods study involved two phases targeting two different participant groups: the first was an initial quantitative phase with healthcare employers (N = 29), which informed a second qualitative phase with DDCs (N = 16). In the first phase, a virtual quantitative survey was administered to gauge healthcare employers’ perspective of DDCs’ wellness during the COVID-19 pandemic, their priorities, and whether support for DDCs was implemented. Here, it was found that not many employers established supports for DDCs due to factors including lack of funding and a lack of awareness about DDCs. The preliminary results from this survey helped to inform and direct the exploration of phase two, which consisted of an approximately 1 h semi-structured qualitative interview with DDCs. For instance, during the interview, DDCs were asked whether they had revealed their double-duty caring status to their employers, what kinds of supports they seek, and questions relating to the employer–DDC relationship. The study protocol received ethical approval from the McMaster Research Ethics Board at McMaster University (MREB#:4874). Although the first stage of this research (the surveys) shaped the second phase (the interviews), this study reports results directly from the second stage of this research: the 16 semi-structured interviews.

### 2.2. Sampling

A purposive sampling approach, specifically through criterion sampling, was utilized to recruit 16 DDCs that fit a predetermined inclusion criterion. DDC participants were recruited based on the following inclusion criteria: (a) working in a caring role in the healthcare or homecare industry in Ontario; (b) providing unpaid care for a family member or friend at least 18 years of age; and (c) having worked at their place of employment for at least one year. For this study, it was important that participants had been working in their paid caring role for at least one year as this would allow for them to have experienced workplace changes due to the pandemic.

Recruitment occurred from January 2021 to March 2021. Various recruitment strategies were employed to obtain a sample that adequately fit the criteria for DDC participants. The healthcare organizations that participated in the quantitative survey from the first phase of this study were contacted to share information about the study and to request recruitment assistance from their staff of healthcare or homecare workers. Various other healthcare organizations across Ontario were also contacted to request recruitment assistance from their staff; however, many organizations at this time were burdened with COVID-19-related challenges and were too busy to assist. Furthermore, the recruitment period coincided with the vaccine rollout for healthcare workers across Ontario, which challenged healthcare organizations’ provision of support in this study given their additional responsibilities.

Under normal circumstances, DDCs would be considered a ‘hard-to-reach’ population given their busy schedules; in fact, unresponsiveness and difficulty targeting DDCs for research has been reported in previous studies [1]. During COVID-19, this was exacerbated, making recruitment even more difficult. For hard-to-reach populations, social media recruitment strategies have proven cost-effective and advantageous given their ability to reach targeted demographics and achieve lead generation [38,39]. A lead-generation campaign, a form of social media advertisement created with the sole purpose of engaging the target audience to fill out a form or complete a short questionnaire, was created on social media channels using targeted paid advertisements.

### 2.3. Procedures

The semi-structured interview guide was designed based on the following factors: (a) the previous literature on DDCs; (b) elements derived from the conceptual framework on double-duty caring by Ward-Griffin, Brown, St-Amant, Sutherland, Martin-Matthews, Keefe, and Kerr [5], and (c) discussion and validation by experts (supervisory committee members). In designing the interview guide, specific questions were created to represent the various domains involved in the double-duty caring conceptual framework, the previous literature on DDCs, and the principles of qualitative interviewing [40]. The purpose of the interviews was to understand how the caring experience changed due to the COVID-19 pandemic; what supports or coping strategies were available and/or being used, and what supports were lacking. The interviews explored these objectives by inquiring about DDCs’ formal and unpaid caring roles, investigating how these roles impact one another, and asking about their self-management or coping strategies.

Due to the provincial social isolation regulations and safety considerations associated with the COVID-19 pandemic, all interviews were conducted virtually through Zoom, a video teleconferencing program. Prior to interviews, each participant was emailed a letter of information that provided an overview of the study, confidentiality details, consent information, the contact information of researchers, ethics details, and compensation details. These details were also verbally summarized prior to the beginning of each interview. The duration of the interview ranged from 40 to 60 min and all participants expressed interest in receiving the study results upon the study’s completion.

### 2.4. Data Analysis

A thematic analysis was employed to determine common patterns in the data; this analysis has been commonly used to report and interpret the experiences of participants [41]. Although mobility was not the primary focus on the interview, it emerged frequently as a theme and is the focus of this paper. For more information on the study design, including the questions included in the interview guide, please see Mehta (2021) [42].

Braun and Clarke’s step-by-step process of thematic analysis was utilized. This approach to thematic analysis utilized the following steps: (a) familiarization with the data; (b) generating initial codes; (c) searching for themes; (d) reviewing themes; (e) defining and naming themes; (f) producing the report [41]. The process of coding and analysing the interview transcripts was completed on NVivo 12.6.1 (QST International, 2021).

Although the sample might be considered low when compared to other qualitative studies focusing on less difficult-to-reach populations, data saturation was met. In other words, interviews were conducted until no new themes were observed [43,44]. This was confirmed through ensuring operational saturation: quantifying the number of new codes being discovered over time and observing a decreasing frequency of new codes from additional interviews [45].

## 3. Results

Table 1 provides a summary of the social and demographic characteristics of DDCs that participated in the interviews. All 16 DDCs were female and primarily held full-time [75.00% (N = 12)] positions in their role as paid carers. During the interviews, ethnicity was not explicitly asked; however, many DDCs brought it up during the discussion of their caring experiences. Among all DDCs interviewed, three (17.24%) were White, three (17.24%) were South Asian, four were East Asian (25.00%), and the ethnicities of 6 individuals was unknown. Half (50.00%, N = 8] of the DDCs reported working in their paid care role for 4–7 years, four (25.00%) had been working for over 8 years, two (12.50%) had been working for 1–3 years, and two had been working for 6 months–1 year. While the relationship to DDCs’ dependents varied, most DDCs’ (56.25%, N = 9) provided unpaid care to their aging parents. Many DDCs (68.75%, N = 11) reported living with their dependents and three (18.75%) individuals reported living with them only during COVID-19, where support was increasingly needed. During COVID-19, 8 (50.00%) of the interviewed DDCs reported being unable to obtain assistance with their unpaid caring responsibilities from their support network, two (12.50%) were able to receive assistance, and six (37.50%) reported receiving assistance sometimes (for example, when COVID-19 restrictions allowed).

Across the board, DDCs reported deteriorating work conditions that were characterized by an increase in work expectations, workload, and concerns surrounding safety during the COVID-19 pandemic. These challenges contributed to high stress levels, emotional strain, and fears of safety, and many DCCs reached the point of burnout. A few participants were visibly crying, angry, or expressing frustration during the interviews.

This paper focuses on the role of mobility in shaping DDCs’ wellbeing, drawing on the Mobility of the Care Economy framework. This results section is organized around the three mobility pathways that were identified as negatively affecting DCCs’ health: Increased Mobility Demands, Forced Decreased Mobility, and Indirect Impacts of Mobility. A visualization of the results can be found in Figure 2.

### 3.1. Increased Mobility Demands

DDCs’ regular schedules were disrupted and they lacked flexibility in work arrangements during the pandemic, which resulted in additional mobility constraints. For instance, short-staffed workplaces meant that staff members were assigned to more hours, more shifts, and extra clients per shift. Those who were able to take on this additional work expressed additional physical and mental strain. These additional shifts also inherently resulted in greater trips to and from work, which incurred spatiotemporal constraints.

As another example, an increase in safety protocols, and specifically COVID testing requirements, resulted in not only additional labour, but additional logistics regarding traveling to and from testing centres. As one participant details:


*The having to go for COVID tests every single week itself is so strenuous. For my work place, they expect us to be tested every week and I know it might not be like that everywhere but the testing isn’t even at the work office. So, the COVID testing is so hard to fit into my schedule, to travel to, make time for, it’s a lot of work.*
(DDC4)

While these additional safety measures were put in place for good reasons, they were not implemented to minimise DDCs’ mobility constraints. As DDC4 states, additional trips to COVID testing centres, centres that were not even conveniently located “at the work office”, led to spatiotemporal constraints that reduced caregivers’ wellbeing.

The two examples above explore increased mobility needs due to paid labour; however, many DDCs also experienced increased mobility needs resulting in spatiotemporal strain due to unpaid care. Many DDCs’ usual supports were no longer available during the pandemic, or they had reduced access to these supports due to lockdown restrictions, which resulted in additional trips being made to access necessities like food, medicine, and other supports. For instance, a DDC explained that her caring expectations rose when supports or resources to help with unpaid caring roles diminished due to COVID restrictions:


*I try not to bring my work home or take my home at work and I used to keep those two different. But as of late […] I’m so exhausted mentally and physically from the hectic work environment. Then to think, oh my god, I need to go help my parents too. […] taking care of my parents got hectic. They now need me to get meds, groceries, clean their place because nobody can come now. At times, I’m not with it and don’t want to go to provide the care that they need […] because I’m just physically tired like my muscles are sore and I’m absolutely burnt out.*
(DDC12)

Due to the new COVID-19 regulations, DDCs lost much of the support that helped them to manage their caring responsibilities; for example, almost all DDCs participants described that the loss of their support network made double-duty caring very emotionally difficult. This was due to DDCs no longer having anyone to provide them with emotional support or assist with caring duties for their dependents. This also resulted in additional mobility expectations; in the example above, DDC12 had to take on the additional mobility of care trips, including those “to get meds, groceries, clean their place”.

In all examples, these increased mobility constraints, whether due to paid or unpaid care labour, were rooted in the increased expectations of DDCs’ employers, i.e., expecting DCCs to pick up other staff member’s shifts, to conduct additional safety protocols, or to conduct trips in an exhausted state to respond to dependents’ needs. In many cases, employers did not reimburse costs of travel, even when additional trips had to be conducted. These increased expectations, combined with a lack of additional resources, left DDCs with additional spatiotemporal constraints, which negatively impacted their wellbeing.

### 3.2. Forced Decreased Mobility

The increased expectations regarding paid and unpaid labour due to COVID-19 did not always result in additional mobility requirements. In many cases, DDCs experienced decreased mobility and spatiotemporal constraints; however, these decreased constraints still negatively affected workers’ wellbeing. Prior to the pandemic, DCCs often relied on services like daycare and homecare as supportive strategies in managing their unpaid caring responsibilities. These institutions that support DDCs’ unpaid care duties, however, were also impacted by the pandemic, resulting in a shifting mobility constraint landscape.

Provincial efforts to limit the spread of COVID-19 led to the closure and limited accessibility of numerous essential or community-based services which played a significant role in supporting the care of vulnerable dependents. Consequently, DDCs could not access supportive services such as homecare, respite, and daycare services and had to sacrifice more of their time and effort to compensate for this. For example, there was scarce availability of homecare services during the pandemic, which meant that DDCs had to overcompensate the amount of care they provided for their dependents, oftentimes reducing their hours at work. While this reduced their spatiotemporal constraints, their reduced mobility still negatively influenced their wellbeing due to the increased care expectations.

For DDCs that care for dependents with chronic conditions, homecare was also an essential service that many depended upon to help with their caring role. Without homecare services, DDCs who had the primary responsibility of being the unpaid carer in the household were forced to stay home from work and lose a day’s income. One DDC, who also bore the pressure of being the family’s sole earner, suffered the drastic economic costs of losing homecare services for her bed-bound husband:


*“I used to depend on home care to help me with my husband. If they’re not coming, I stay home from work and help him. You don’t get paid for that…So I lose the day’s pay, and I’m the only one working in the family.”*
(DDC1)

Furthermore, all DDCs interviewed were parents who relied on daycare or school services to balance their time between paid work and unpaid caring. When schooling became remote, their children were always home. Although not having to drop off and pick up children from school eased the mobility burden for DDCs, their wellbeing was still negatively affected as staying home to parent and homeschool their children resulted in financial losses.

Daycares were more unpredictable than schools. With COVID-19, daycare operations were heavily reduced and had stricter safety regulations, resulting in children being sent home frequently and unexpectedly due to the suspected sickness of any child. This was incredibly stressful for DDCs as they had to suddenly leave work to care for their children. This was not only an additional mobility constraint for DDCs, but also led to many experiencing recurring fears that their child could have been infected at that time, as the following DDC demonstrates:


*My children are also getting sent home from daycare quite frequently […] the daycare rule is a negative COVID test, doctor’s note or 10 days out of daycare…the other day, I was at work and had to leave in a very big rush because of having to get my children who were being sent home from daycare. As my daughter has a runny nose from allergies, she’s sent home from daycare. COVID’s put a lot of strain […] I try to have backup arrangements when I have to work, however, a lot of the times, I can’t end up getting anybody and it always pushes me out of work.*
(DDC5)

Other than the emotional strain, participants also discussed the financial impact that forced reduced mobility had. For example, one DDC reported increased financial stress from factors like having to pick up her kids from daycare, as they kept being sent home due to suspected sickness:


*[…] financially, it’s stressful because when I’m out of work. So, financially, it’s hard and it’s extremely, extremely stressful because you never know what’s going to happen. […] I never know if they’re going to be out for one day or for ten days. I never know if my parents will have an emergency.*
(DDC5)

For DDCs with multiple dependents to care for, the unpaid responsibilities were even greater, making it difficult for DDCs to manage the same paid work hours as before the pandemic.

Normally, relatives or other members of DDCs’ support network would fill in when daycares were closed; however, due to the social isolation regulations, this was often not an option.

In all three cases, homecare service scarcity, remote schooling, or daycares sending children home frequently and unexpectedly, these unpaid care constraints influenced with DDCs mobility regarding paid employment; they had to reduce their hours and therefore go to work less frequently. While this meant they stayed home—and therefore reduced their mobility and spatiotemporal constraints—it still resulted in decreased wellbeing due to financial loss and the associated increased stress.

### 3.3. Indirect Impacts of Mobility

Other people’s mobility was also identified as influencing DCCs’ wellbeing. Indeed, restrictions during the pandemic meant that many of the people who helped to support those in need of care were no longer able to visit them in person. This occurred amongst the dependents that DDCs cared for in both a paid and unpaid capacity. This reduced support resulted in increased expectations on DCCs. Specifically, almost all DDCs shared that it was difficult to cope with the declining wellbeing of their dependents, who were displaying challenging and agitated behaviours while trying to cope with social isolation during the pandemic. This social isolation was indirectly due to mobility; while DDCs were still visiting the people they cared for, few others were able to visit in order to limit the risk of their acquiring the virus. DDCs reported experiencing a variety of negative emotions, such as sadness, helplessness, and frustration, when caring for their dependent(s); DDCs felt that these emotions put strain on the caring relationship. This illustrated that DDCs’ wellbeing is closely impacted by the wellbeing of their dependent(s). All DDCs noted a decline in the wellbeing of their dependent(s) due to COVID-19; DDCs also felt a considerable amount of emotional strain and pressure upon themselves to rectify their mood.

For example, one DDC, who cared for her mother who was suffering from cancer, expressed that her own wellbeing was often closely tied to that of her mother’s. She verbalized a sense of emotional strain and feelings of helplessness in being unable to support her mother:


*My mom has four sisters, so they would always visit and they’re a really big emotional support system for her too, so now they can’t really come and be there for her. Last week, she said to me “You know, I’m a little bit depressed” and that made me so sad, because I can’t fix that either. I wish I could fix a lot for her but I can’t. When my mom doesn’t have a good day I don’t have a good day.*
(DDC10)

Of note, this decline in DDC wellbeing was directly related to the reduced mobility of her aunts, who could no longer visit her mother. This resulted in reduced emotional support being available to her dependent, her mother, and increased feelings of isolation.

Another DDC recalled a similar experience with an elderly friend she cared for, describing how social isolation amplified feelings of loneliness and seclusion. In the hope of lifting her mood, she attempted to provide her with dependent special care and explore additional ways to care in order to improve her wellbeing:


*The pandemic’s taken such a big toll on her. […] she’s been locked indoors for almost a year. She hasn’t met her friends, she used to go out for social gatherings with them. Like, she would meet up with friends for probably a coffee before. Most days she is in front of the TV all alone […] It makes me really sad to see her so depressed […] I force her to get her nails done […] I try to wash her hair and style it because extra grooming is something that puts a smile on her face.*
(DDC13)

The social isolation orders and the lack of socialization due to the pandemic also caused some dependents to demonstrate agitated and destructive behaviours that were difficult for DDCs to manage. It was very stressful trying to manage dependents’ agitation, and it was emotionally exhausting for DDCs who worked constantly to improve their mood. DDCs expressed being under exacerbated emotional circumstances and felt helpless, not knowing how to help their dependents. The growing mental health challenges of dependents was a dominant theme of discussion; these challenges caused DDCs to take on greater responsibilities in providing companionship and emotional support to manage their dependents’ wellbeing. Another DDC, who cared for her bed-bound cousin, stated that while she wants to provide companionship, the added responsibilities are sometimes hard to manage:


*For her, it is hard too […] She’s very depressed some days and cries to me…it make me very sad for her and make me want to come and stay with her…but I cant because I have work and my son to look after too. There’s not enough time in my day to stay longer with her…I really feel bad for her.*
(DDC7)

Similarly, at work, DDCs had to manage the wellbeing of clients who required much more physical assistance and emotional support in their activities of daily living (ADL) during the COVID-19 pandemic. The increased emphasis on mental health management and the provision of emotional support in DDCs’ paid and unpaid caring responsibilities was not something DDCs were originally trained or prepared for. At work, managing clients’ wellbeing exacerbated DDCs’ caring work and, in combination with intensified unpaid caring burdens, contributed to frequent instances of burnout:

*The residents are much sicker, agitated and confused being isolated in their rooms […] A patient who was previously able to use their walker to ambulate, they’re now weaker because they’re not walking around as much. We’re not doing recreational activities outdoors either so…their wellbeing is going down and that means that they’re far more dependent on us. We go home so physically exhausted that combined with the emotional exhaustion and stress from worrying and caregiving for your family member and not even being able to help the residents to the best you can is just too much. It’s just such a good cocktail for burnout. So yes, this last year was burnout galore.*
(DDC15)

While some DDCs’ clients showed vigilance, others suffered from poor mental health symptoms because of the provincial social isolation protocols which restrict family visiting in healthcare settings. Given that many of the interviewed DDCs worked with an at-risk populations (aging and/or immunocompromised clients) in a healthcare facility, the strict COVID-19 social isolation regulations prevented clients from closely interacting with others in the facility and being able to see their families for long periods of time. The lack of socialization and companionship makes clients lonely, tense, and very agitated, which makes ADL caring tasks very strenuous for healthcare staff. Previously, staff relied on the support of volunteers, recreation programs, or family to maintain their clients’ wellbeing and provide companionship; however, due to COVID-19 safety protocols, this support was no longer available. Instead, the socialization and maintenance of clients’ wellbeing was an added responsibility that DDCs must shoulder at work:


*Usually, volunteers or family come to help with companionship. Now, we […], talk to them, socialize and so we never get breaks from taking care of them. Community centres are closed so […], it takes a lot of time and effort doing online research from us to find Zoom activities […] It takes a lot of time to calm them down and sometimes it is so stressful because we are not taught what to do in this case. […] In the past, they looked forward to like birthday parties, getting with their families, which cannot happen so it takes a toll on them.*
(DDC15)

Not only did the added responsibility of managing clients’ mental health create stress for paid carers, but this was also not a task that workplaces provided training or preparation for. As one DDC mentions, “*You’re basically working to physically distance which is hard because they all have mental health issues now so socialization is what they need”* (DDC14). While DDCs empathized with clients’ need for socialization, they were also bound by safety protocols that barred them from providing this care.

Indirect mobility also shaped DDCs’ wellbeing throughout the pandemic; the reduced ability of other carers to visit those in need of care resulted in additional strain on DDCs.

## 4. Implications for Practice

This study highlights how mobility demands, whether direct or indirect, shape DDCs’ wellbeing. However, mobility was rarely a consideration for employers. This section explores the ways in which employers could have accommodated DDCs’ mobility needs during the pandemic. Indeed, many participants identified workplace policies that eased their burden and workload. Looking across responses, supportive work environments provided the following: flexible work hours, frequent brief check-ins, access to workplace counselling services, peer-to-peer support groups, and self-care (journaling, wellness apps, meditating, walks, reading, puzzles, etc.). These policies are all connected to mobility, either directly or indirectly, as described below.

Flexible work hours relate directly to mobility in that they allow for more flexible movement between paid and unpaid care commitments. Indeed, many DDCs wished for more flexible scheduling and a greater consideration for employees with unpaid caring responsibilities. One DDC, who had revealed her DDC needs, describes that the scheduling struggles were incredibly stressful and often caused her to miss medical appointments:


*I was letting them know I can only do 8 am–4 pm. When I had doctor appointments, they would change my hours without my knowledge and it would make me miss doctors appointments. […] It’s so frustrating because they’re constantly calling and asking, ‘hey can you work at 5:00 AM?[…] I would like them to be more attentive to scheduling. I have back injury […] and I have a doctor’s appointment every single week and there’s been several occasions where they changed my shift without notifying me and I’d miss my appointments because I would have to be at work.*
(DDC5)

Weekly check-ins by employers were found to help with the logistics of the mobility of the care economy, for example, by coordinating paid and unpaid care commitments. When DDCs needed to be tested for COVID-19 for work, they were able to also take a family member to an appointment. As one DDC noted:


*So the weekly check-ins are helpful, because then you can discuss things that are changing, and then you feel kind of better at the end of the conversation. And they can let us know that like I am assigned to get tested in 4 weeks on that date so we can plan ahead and with my mom’s weekly appointments, that’s good for me to know.*
(DDC11)

Peer-to peer support was greatly valued by many DDCs involved in this study, especially with other DDCs. This strategy provided emotional support and helped DDCs feel that they are not alone in their experiences. Of note, counselling, peer-to-peer (or family) support, and self-care strategies are also indirectly related to mobility, as interviewees noted that these are often accomplished through technologies (by phone, group messaging, etc.) to reduce the mobility requirements.

## 5. Discussion

This paper contributes to the COVID-19 travel behaviour [46,47,48,49] literature by exploring the mobility experiences of DDCs during the COVID pandemic. During the pandemic, the number of trips taken globally decreased dramatically because of safety restrictions (e.g., lockdowns), and social distancing practices [46,47,50]. Studies show substantial drops in travel times during the COVID-19 pandemic, regardless of age group and gender, for the general population [47]. Differences across populations, however, were observed. For instance, socio-economic variations in travel behaviour were identified [51]. Further, essential workers, including those working in healthcare, were the expectation. They travelled more than the general population during the pandemic as they continued to work outside of the home and often had increased work responsibilities [48]. This paper provides additional nuance to these findings. DDCs’ travel behaviour was at times greater and at other times reduced during the COVID pandemic due to a complex combination of resources, expectations, and spatiotemporal constraints.

This study also extends the mobility of care literature [29,30,31,32,52,53] by showing how mobility for paid care and mobility for unpaid care shape each other. Indeed, most of the mobility of care literature focuses solely on unpaid care mobilities [29,30,32]. This study highlights the importance of considering care mobilities for unpaid and paid labour. According to the Mobility of the Care Economy framework, the caregiving interface is dependent on DDCs’ paid and unpaid caring responsibilities, with mobility in these two realms being directly impacted by resources, expectations, and spatiotemporal constraints. The COVID pandemic influenced all these factors; DDCs were working with fewer resources (e.g., the closure of daycares, schools, and day centres, and reduced help from other carers), with increased expectations (e.g., additional working shifts, increased safety requirements), and additional spatiotemporal constraints (e.g., lockdowns). These factors influence DDCs mobility in many ways, at times by increasing the amount of travel required and at other times by restricting travel, and other people’s mobility was impacted in ways which impacted DDCs. In studying this, this paper is the second to directly examine the role of mobility of care on wellbeing. As was the case in past work [33], negative associations were found between this travel and wellbeing.

Looking across the COVID travel behaviour and mobility of care literature, the pandemic has provided evidence that care work, whether paid or unpaid, has social and economic outcomes for society.

This study is not without limitations. The geographic focus is on one region in Canada: Southern Ontario. It is possible that the results do not translate to other Canadian cities, other regions, or other global contexts. This research also only focused on female DDCs. Further research that includes both male and female DCCs could uncover the gendered nature of mobility of care in double-duty caregiving. This future research would require a larger sample to tease out the variations expected across genders. Finally, it is possible that some of the mobility constraints identified in this study only apply during the COVID-19 pandemic. While these results can help inform policies during future pandemics, some results may be less applicable during non-pandemic conditions.

Additional research on the mobility of DDCs post-pandemic is needed given that many of them are managing several part-time paid positions while simultaneously providing unpaid care to dependent(s) with complicated care needs. The objective of this research would best be focused on sustaining and improving wellbeing to prevent burnout, which is a real threat for DDCs. Further, all DDCs in this study were female and most were visible minorities (all but four that did not disclose their ethnicity/race). This is perhaps unsurprising as, across OECD countries, women took on the brunt of the caregiving responsibilities during COVID, which resulted in labour market penalties (e.g., job loss) and financial stress [54]. Women have also consistently been found to have lower access to personal vehicles and to be more transit-reliant than men [55]. Further, households headed by visible minorities are less likely to have access to a vehicle as compared to white households [56,57]. Research has also found that public transportation is a concern for caregivers at the intersection of race and immigration status due to discrimination [58]. While the interviews did not focus on this, future work could explore the gendered and racialized facets of care workers’ transport experiences, such as mode choice, access to transport, and experience while mobile. This future work would highlight some of the important equity considerations for the mobility of the care economy.

## 6. Conclusions

This paper answers the following research question: what role did mobility play in shaping DDCs’ health and wellbeing during the COVID pandemic? The Mobility of the Care Economy Framework was followed. Mobility of care, both for paid and unpaid care, was found to negatively shape DDCs’ wellbeing. Three mobility pathways were identified as negatively shaping DCCs’ wellbeing during the COVID-19 pandemic. First, certain protocols resulted in increased demands, including additional testing requirements and the need to take on additional hours due to labour shortages. Second, other policies, including the closure of schools and daycares for children and recreational and day programs for adult and elderly dependents, resulted in reduced mobility demands for DDCs. These, nonetheless, decreased their wellbeing as they tended to result in reduced working hours and increased financial stress. Third, pandemic restrictions severely limited the mobility of other sources of unpaid care support (i.e., from family, friends, and neighbours). This resulted in additional strain on DDCs. At times, this also increased the amount of mobility of care trips that DDCs had to make to adult/elderly dependents who were not living in the DDC’s household; these trips included grocery and/or pharmacy drop-offs, health checks, socio-emotional care, and/or pick up/drop off for health appointments.

These three mobility pathways all resulted in reduced health outcomes amongst DDCs, highlighting the importance of considering mobility considerations. To counter this, this research identified policies that could improve DCCs’ wellbeing. First, flexible work hours could greatly improve DDCs’ wellbeing by enabling them to meet all their complex mobility demands. Second, weekly check-ins would provide DCCs with a space to describe the logistical challenges of their mobility of the care economy needs and a space to resolve these needs with their employers. Third, counselling, peer-to-peer support, and self-care strategies that can be provided remotely could improve DDCs’ wellbeing without causing additional mobility constraints. Fourth, peer-to-peer support, particularly with people in similar situations, can benefit DDCs and potentially other care-related employees. As depicted in the framework, supports for DDCs go a long way in improving mobility in unpaid caring work, as well as allowing for a balancing of responsibilities.

## Figures and Tables

**Figure 1 ijerph-21-00730-f001:**
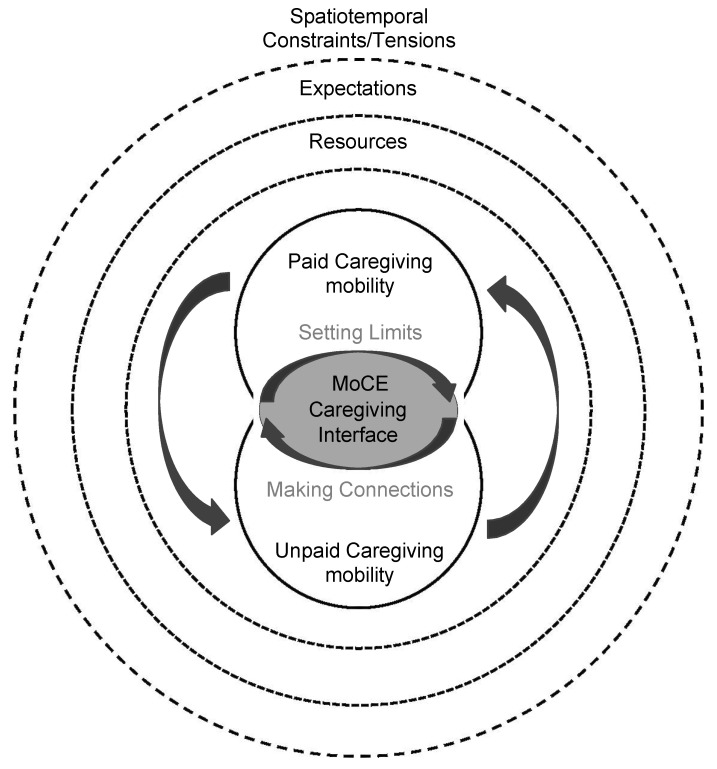
Modified ‘Mobility of the Care Economy Framework’ [Williams et al. (forthcoming) [35]].

**Figure 2 ijerph-21-00730-f002:**
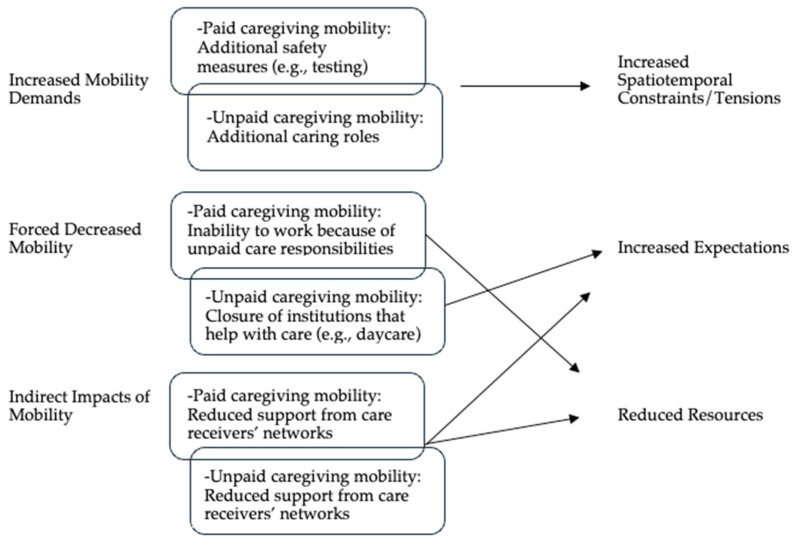
Mobility of the care economy’s Impact on DDCs’ health and wellbeing.

**Table 1 ijerph-21-00730-t001:** Social and demographic characteristics of DDCs.

Characteristics	N	%
**Employment**		
Part-time	1	6.25
Full-time	12	70.00
Part-time, working full-time hours	3	18.75
**Ethnicity**		
White	3	18.75
South Asian (East Indian, Pakistani, Sri Lankan)	3	18.75
East Asian (Chinese, Japanese, Republic of Korea)	4	25.00
Unknown	6	37.50
**Gender**		
Man	0	0
Woman	16	100.00
**Where are you currently working?**		
Hospital	4	25.00
Long-term care or nursing home	5	31.25
Homecare	6	37.50
Outpatient facility	1	6.25
**How long have you worked in this role?**		
6 months–1 year	2	12.50
1–3 years	2	12.50
4–7 years	8	50.00
8+ years	4	25.00
**Relationship with Dependent**		
Parent(s)	9	56.25
Grandparent(s)	2	12.50
Spouse	1	6.25
Uncle	1	6.25
Cousin	1	6.25
Friend	2	12.50
**Living with Dependent**		
Yes	11	68.75
No	2	12.50
Only during COVID-19	3	18.75
**Assistance from Support Network during COVID-19**		
Yes	2	12.50
No	8	50.00
Sometimes	6	37.50

## Data Availability

The datasets presented in this article are not readily available because of the anonymity clauses inherent in qualitative research. For more information, please see [24]. Requests to access the datasets should be directed to Sakshi Mehta.

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
