# Peer review of "Double-Duty Carers’ Health and Wellbeing during COVID-19: Exploring the Role of Mobility of the Care Economy in Southern Ontario, Canada"

_ijerph, 2024, doi:10.3390/ijerph21060730_

Round 1
Reviewer 1 Report
Comments and Suggestions for Authors
This is a useful exploratory study. I provides several unexpected findings. It is easy to overlook the totality of effects in a very complex system including both job and home components
Author Response
Thank you for taking the time to review our manuscript. We are delighted you find it useful and insightful in its unexpected findings.
Reviewer 2 Report
Comments and Suggestions for Authors
The article fits well within the theme and scope of the journal. It presents important aspects of healthcare during the COVID-19 pandemic. To improve the quality of the text, certain changes are suggested:
Article title: If the presented research results pertain solely to studies conducted in Canada, then the title of the article should include this limitation. Otherwise, the article's title may mislead readers and suggest that the research has a global character. In its current form, the title of the article is significantly broader than the research presented within it.
It is stated, "Unpaid caring work provides an annual contribution of $25 billion to the economy [5] and serves as the backbone of the healthcare system…" Is this referring to costs in a global sense or in specific countries or a country? (line: 42-43). This issue should be clearly stated.
Lines: 492-495. The fragment starting with the words, "This paper contributes to the emergent mobility of care…" adds little to the content of the discussion. This fragment should be removed.
The manuscript lacks a Limitations section. The limitations of this article are evident: the research is limited to one country (specifically to Hamilton), firstly, and secondly, it is based solely on 29 interviews. Therefore, another question arises as to how representative the presented research can be in a broader context, considering other parts of Canada?
Author Response
Thank you for positive remarks and your insightful suggestions to improve the manuscript. All have been addressed and are detailed below (in Italics after your original comment).
Article title: If the presented research results pertain solely to studies conducted in Canada, then the title of the article should include this limitation. Otherwise, the article's title may mislead readers and suggest that the research has a global character. In its current form, the title of the article is significantly broader than the research presented within it.
We have changed the title to reflect where this research took place to: Double Duty Carers’ Health and Wellbeing during COVID-19: Exploring the Role of Mobility of the Care Economy in Southern Ontario, Canada. We have also added the possibility that results do not translate to other global contexts as a limitation in the new limitations section (see 557-565).
It is stated, "Unpaid caring work provides an annual contribution of $25 billion to the economy [5] and serves as the backbone of the healthcare system…" Is this referring to costs in a global sense or in specific countries or a country? (line: 42-43). This issue should be clearly stated.
We are very glad you noted this issue, as we had not realised the old wording make it sound like these two studies had the same geographic context. This has been rectified in the revised version which now reads: “Unpaid caring work provides an annual contribution of $25 billion to the Canadian economy [5]. Globally, this unpaid care serves as the backbone of the healthcare system [6].” (please see lines 44-45).
Lines: 492-495. The fragment starting with the words, "This paper contributes to the emergent mobility of care…" adds little to the content of the discussion. This fragment should be removed.
This fragment has been removed (see line 522). The mobility of care literature is mentioned and cited in the following paragraph, which is more appropriate (lines 536-551). Thank you for noting this.
The manuscript lacks a Limitations section. The limitations of this article are evident: the research is limited to one country (specifically to Hamilton), firstly, and secondly, it is based solely on 29 interviews. Therefore, another question arises as to how representative the presented research can be in a broader context, considering other parts of Canada?
This is an excellent suggestion, and one that was noted by other reviewers. A limitation paragraph has been added. It outlines the limitations you note about representativeness of the results to other geographic contexts, as well as other limitations noted by the other reviewers.
We disagree, however, that the sample size of the research is a limitation. Qualitative research does not adhere to the positivist criteria of rigour that quantitative studies do. Rather than focus on sample size, qualitative researchers must focus on quality and trustworthiness. We are confident of the quality of the data as saturation was reached with the 16 interviews. In fact, some research states that a sample can be too large in qualitative research (e.g., see Sandowski (1995) as it does not permit the deep analysis that is the raison d’être of qualitative inquiry. Also, it depends on what type of qualitative work is being done -as well as the quality of the data, the scope of the study, and the qualitative method used. Sample sizes of 5-10 is common in phenomenology whereas a larger sample of 20-30 is necessary in grounded theory. Because our research was very focussed in its scope with respect to health and wellbeing of female DDCs and their mobility to care, our sample of 16 falls within the proposed range. We do, instead call for future research that extends this present study which would require larger sample sizes.
Please see lines 557-565. And thank you for this suggestion. We believe the new limitations section adds to the manuscript.
Reviewer 3 Report
Comments and Suggestions for Authors
Dear authors, thank you for the opportunity to review your research.
The strengths of the study are the uniqueness of the empirical object, the originality of the model, the qualitative approach to the research, and the practical orientation of the research.
At the same time, there are comments and recommendations for improving the work:
1. The introduction reveals the relevance of the study, indicates its purpose, but the authors do not present the research hypothesis.
2. A big question regarding the study sample and its sufficiency. In the summary of the work, the authors indicate that 29 interviews were conducted. And the article states that there were 29 participants at the first stage of the study, at the second stage the sample was only 16 people. This is a discrepancy. And the justification for the sufficiency of 16 people to conduct this qualitative study is not presented. Why was it impossible to increase it?
3. The materials and methods also mention interview questions, and then analyze the interviews in three directions regarding mobility. However, the authors do not indicate whether there were any questions about mobility? What other questions were answered besides mobility?
4. The authors write in the materials and methods that the interview was analyzed by content analysis, coding and distribution of answers into categories was carried out, while the text of the work does not provide a single table summarizing the results of this coding. This needs to be supplemented so that it is clear on what basis the qualitative descriptions are given, is everything taken into account in the description?
5. The Materials and Methods section itself must be structured by subsections: methods, sampling, procedure, data analysis
6. In the description of materials and methods, the authors say about the first stage of the study that its results were taken into account to develop interviews for the second stage. Should we elaborate on this, how was this taken into account? What was determined in the first stage of the study?
7. Section Discussion of the results is very brief; little analysis and correlation with the results of studies by other authors is provided.
8. No description of study limitations
9. The conclusion is very voluminous, it is necessary to highlight it more clearly and succinctly, according to the purpose and hypothesis of the study.
In this connection, the article can be recommended for publication after significant revision.
Best wishes, reviewer
Author Response
Thank you for reviewing our work. We are delighted to hear your positive remarks and appreciate the ways in which you suggest we improve this manuscript, which are all address below. For clarity, we have left your original comments, and responded to them individually in Italics.
- The introduction reveals the relevance of the study, indicates its purpose, but the authors do not present the research hypothesis.
As this is a qualitative study, it does not aim to test a direct hypothesis. Rather, it aims to answer a research question. Your comment helped us realize that we had not stated this research question clearly and have therefore added this on lines 120-121: “In doing so, this paper aims to answer the research question: what role does mobility play in shaping DDC’s health and well-being?”. We have also reiterated this research question in the Conclusion. Given that this reads like a hypothesis, we are hopeful this responds to your excellent comment.
- A big question regarding the study sample and its sufficiency. In the summary of the work, the authors indicate that 29 interviews were conducted. And the article states that there were 29 participants at the first stage of the study, at the second stage the sample was only 16 people. This is a discrepancy. And the justification for the sufficiency of 16 people to conduct this qualitative study is not presented. Why was it impossible to increase it?
This manuscript relies on the 16 interviews that were conducted. Your comment helped us realize that this was not clear in the original manuscript. This has been clarified in the revised manuscript on lines 146-148: “Though the first stage of this research (the surveys) shaped the second phase (the interviews), this study reports results directly from the second stage of this research: the 16 semi-structured interviews.”
Though 16 interviews may seem like a small sample, we are confident the data are sufficient. First, DDCs are considered a “hard to reach” population given their immense time constraints. The difficulty of recruiting DDCs has been reported in other studies. This difficulty was likely only exacerbated during COVID-19 when DDCs were even more busy, which coincided with the recruitment of this research. This has been noted on lines 170-173: “Under normal circumstances, DDCs would be considered a ‘hard-to-reach’ population given their busy schedules; in fact, unresponsiveness and difficulty targeting DDCs for research has been reported in previous studies [1]. During COVID-19, this was exacerbated, making recruitment even more difficult.”.
Importantly, our confidence in the data is because saturation was reached: data collection only stopped when no new themes emerged. This is a common tool to ensure rigor is qualitative research, something we have made clear in the revised version. Further, the research took one additional step and conducted analyses to do operational saturation, all of which is now noted on lines 210-215 “Though the sample might be considered low when compared to other qualitative studies focusing on less difficult to reach populations, data saturation was met. In other words, interviews were conducted until no new themes were observed [26, 27]. This was confirmed through ensuring operational saturation: quantifying the number of new codes being discovered over time and observing a decreasing frequency of new codes from additional interviews [28].”
- The materials and methods also mention interview questions, and then analyze the interviews in three directions regarding mobility. However, the authors do not indicate whether there were any questions about mobility? What other questions were answered besides mobility?
This information has been added in the revised manuscript. Please see lines 201-204: “Though mobility was not the primary focus on the interview, it emerged frequently as a theme, and is the focus of this paper. For more information on the study design, including the questions of the interview guide, please see Mehta (2021) [30].”
- The authors write in the materials and methods that the interview was analyzed by content analysis, coding and distribution of answers into categories was carried out, while the text of the work does not provide a single table summarizing the results of this coding. This needs to be supplemented so that it is clear on what basis the qualitative descriptions are given, is everything taken into account in the description?
Thank you for this comment. This manuscript focuses on DDCs mobility, and therefore reports the results related to that theme. This has been clarified in the manuscript, and a citation has been added for those who wish to read more on the study design, interview guide, and coding process. Please see lines 200-204: “Thematic analysis was employed to determine common patterns in the data and has been commonly used to report and interpret the experiences of participants [41]. Though mobility was not the primary focus on the interview, it emerged frequently as a theme, and is the focus of this paper. For more information on the study design, including the questions of the interview guide, please see Mehta (2021) [42].”
- The Materials and Methods section itself must be structured by subsections: methods, sampling, procedure, data analysis
This excellent suggestion has been incorporated into the manuscript. We thank you as we believe it makes Section 2 more readable.
- In the description of materials and methods, the authors say about the first stage of the study that its results were taken into account to develop interviews for the second stage. Should we elaborate on this, how was this taken into account? What was determined in the first stage of the study?
We have added some additional information on lines 133-144: “This mixed-methods study involved two phases targeted at two different participant groups: an initial quantitative phase with healthcare employers (n = 29) which informed a second qualitative phase with DDCs (n = 16). In the first phase, a virtual quantitative survey was administered to gauge healthcare employers’ perspective of DDCs’ wellness during the COVID-19 pandemic, their priorities, and whether support for DDCs was implemented. Here, it was found that not many employers established supports for DDCs due to factors including lack of funding and lack of awareness about DDCs. The pre-liminary results from this survey helped to inform and direct the exploration of phase two, which consisted of an approximately 1-hour semi-structured qualitative interview with DDCs. For instance, during the interview DDCs were asked whether they had reveled their double-duty caring status to their employers, what kinds of supports they seek, and questions relating to the employer-DDC relationship.”
We also cite another work with more detail on the methods for those interested in this level of information. Please see lines 201-204: For more information on the study design, including the questions of the interview guide, please see Mehta (2021) [30].”
- Section Discussion of the results is very brief; little analysis and correlation with the results of studies by other authors is provided.
We have revised the Discussion is response to your and other reviewers’ comments. It now includes ample more literature, as well as a new Limitations section and a section outlining future research directions. Please see pages 12-13.
- No description of study limitations
In response to this and other similar suggestions from the other reviewers, a limitations section has been added. Please see lines 557-565: “This study is not without limitations. The geographic focus is on one region in Canada: Southern Ontario. It is possible that the results do not translate to other Canadian cities, regions, or other Global contexts. This research also only focused on female DDCs. Further research that includes both male and female DCCs could uncover the gendered nature of mobility of care in double duty caregiving. This future research would require a larger sample to tease out the variations expected across genders. Finally, it is possible that some of the mobility constraints identified in this study only apply during the COVID-19 pandemic. While these results can help inform policies during future pan-demics, some results may be less applicable during non-pandemic conditions.”
- The conclusion is very voluminous, it is necessary to highlight it more clearly and succinctly, according to the purpose and hypothesis of the study.
The revised conclusion now has two paragraphs: one that summarizes the findings, and one that outlines future research direction. The first one now more clearly responds to the research question by stating: “This paper answers the research questions: what role does mobility play in shaping DDC’s health and well-being during the COVID pandemic? following the Mobility of the Care Economy Framework. Mobility of care, both for paid and unpaid care, is found to negatively shape DDC’s well-being. Please see lines 587-590.
Reviewer 4 Report
Comments and Suggestions for Authors
The article entitled 'Double Duty Carers' Health and Wellbeing during COVID-19: Exploring the Role of Mobility of the Care Economy' addresses an interesting research topic. The authors oriented their attention to the sphere of Double Duty Carers (DDC) and the dimension of high levels of care burden. In order to determine how to best support their health and well-being, the well-being of DDCs during the COVID-19 pandemic was examined, focusing on the factor of mobility limitations.
The executive summary defines the scope of the study and presents the main findings. It also points out the literature gap in response to which the study was written, to emphasise the importance of the research and the novelty of the study. The abstract should also indicate the methodology of the study in a little more detail, which needs to be completed.
The introductory section presents the background of the study, with reference to the literature. It also indicates the direction of the research and its reference. The introductory section should emphasise the importance of the research during the pandemic period. The current need to present the results of research from two years ago should also be explained.
The study lacks an isolated literature review section, so this aspect should either be reinforced in the introduction or a separate section should be introduced, where the review of existing literature findings in the area under study will generally be more extensive. The above will help to develop the layout of the references, which number only 37 items.
Section 2 presents the methodology. It discusses the mode, scope and period of the research, and presents the sample. The research is for the year 2021 (the title also highlights the time of the pandemic as the research reference). I believe that this is quite a distant time, so the current relevance of these results needs additional explanation in the text.
Section 3 presents the findings of the research. It discusses the findings in terms of social and demographic characteristics of DDCs, and further cites findings supported by source statements.
I believe that the research findings should be strengthened from a scientific perspective. I believe that a qualitative analysis should be done, statistical instruments should be applied. Scientifically establish the impact of the social and demographic characteristic(s) on the specific outcome of the findings identified for the purpose of the study.
The discussion part should be strengthened. It is important to have a discussion with the literature in terms of the findings to date versus the research results obtained.
Literature - as already indicated needs to be strengthened. The notation of the bibliography should be refined according to MDPI standards.
To summarise.
The topic is interesting, but the research refers to a distant measurement in time (2021), which requires appropriate explanations and justification of the topicality of the topic (summary, methodology).
The background of the research needs a stronger reference to the literature. I recommend developing the introductory section or adding a separate 'literature review' section to more fully explain the research problem and relate it to the literature, strengthening the number of items in the bibliography.
The results of the research should be strengthened on the scientific side - a qualitative study should be performed, statistical tools for the findings should be applied, etc.
The discussion with the literature and the implications should be strengthened.
Literature needs to be strengthened (number of references). Required bibliographic notation according to MDPI standards.
Author Response
Thank you for taking the time to review our manuscript, and for providing this accurate summary. We have responded to all of your comments, and detail each edit made below (in Italics, and under your original comment).
The executive summary defines the scope of the study and presents the main findings. It also points out the literature gap in response to which the study was written, to emphasise the importance of the research and the novelty of the study. The abstract should also indicate the methodology of the study in a little more detail, which needs to be completed.
The abstract now includes more detail on the methodology, please see lines 16-18 “Following the Mobility of the Care Economy framework and a qualitative research design, it does so through a thematic analysis of 16 semi-structured interviews with female DDCs in Southern Ontario, Canada. Once data saturation was reached, three mobility pathways during the pandemic are identified, all of which negatively affected DDCs wellbeing.”
The introductory section presents the background of the study, with reference to the literature. It also indicates the direction of the research and its reference. The introductory section should emphasise the importance of the research during the pandemic period. The current need to present the results of research from two years ago should also be explained.
This is a good point. Thank you for raising it. We believe that research done over COVID-19 is still important for two reasons. First, it is likely that we will experience a similar event again, be it another pandemic or a different natural disaster, climatic disaster, or political event with similar impacts. Learning from COVID-19 can inform how to best manager future events. Second, this study adds a unique perspective to the literature on health and wellbeing by focusing on the role of mobility for both paid and unpaid work. Mobility very likely affects health and wellbeing during non-pandemic times. We hope this insight will help shape policies that reduce the negative mobility impacts of DCCs and other carers.
We have added these justifications in the Introduction and the limitations sections of the revised manuscript. Pleases see lines 122-127 in the Introduction:
“Though many aspects of the COVID-19 pandemic no longer impact our daily lives, focusing on this time can still offer significant insight. First, were another pandemic to occur in the future, or a different catastrophe with similar impacts, results from this work could inform how best to manage future events. Further, the focus on mobility and its impacts on wellbeing in the context of DCCs in unique, and likely important to consider when crafting carer-friendly policies during non-pandemic times. “
And 562-565 in the Discussion:
“Finally, it is possible that some of the mobility constraints identified in this study only apply during the COVID-19 pandemic. While these results can help inform policies during future pandemics, some results may be less applicable during non-pandemic conditions.”
The study lacks an isolated literature review section, so this aspect should either be reinforced in the introduction or a separate section should be introduced, where the review of existing literature findings in the area under study will generally be more extensive. The above will help to develop the layout of the references, which number only 37 items.
We did not include a separate literature review because we followed the section guidelines put forward by this journal, which states that research manuscripts should have the following sections: “Research manuscript sections: Introduction, Materials and Methods, Results, Discussion, Conclusions.” Please see the IJERPH guide for authors for more information: https://www.mdpi.com/journal/ijerph/instructions#preparation.
We agree that the review could be more extensive, and have added more references. The total number of articles cited is now 59. Though this has made our manuscript significantly longer, we agree that it strengthens the quality of our work.
Section 2 presents the methodology. It discusses the mode, scope and period of the research, and presents the sample. The research is for the year 2021 (the title also highlights the time of the pandemic as the research reference). I believe that this is quite a distant time, so the current relevance of these results needs additional explanation in the text.
As discussed above, this is an important point. We have justified the importance of this work during this time frame in the introduction (see lines 122-127) and added this as a potential limitation (see lines 562-565).
Section 3 presents the findings of the research. It discusses the findings in terms of social and demographic characteristics of DDCs, and further cites findings supported by source statements.
I believe that the research findings should be strengthened from a scientific perspective. I believe that a qualitative analysis should be done, statistical instruments should be applied. Scientifically establish the impact of the social and demographic characteristic(s) on the specific outcome of the findings identified for the purpose of the study.
Thank you for this comment. We have added more writing in the Data Analysis section to strengthen the readers confidence in the findings (please see section 2.4 Data Analysis). We also agree that it would be very interesting to examine the impact of social and demographic characteristics on the outcomes. We believe, however, that the sample makes this impossible. The small sample size, though common in qualitative studies, makes any statistical analysis challenging. Further, as Table 1 demonstrates, only one participant worked part-time, and all participants were women, making a gendered analysis challenging (though this is unsurprising as DDCs tend to be women). Ethnicity would be an interesting variable, but no themes emerged (something we note in the future research directions paragraph). While we think this analysis would be valuable, we believe it outside the scope of this study. Instead, we are hopeful that this paper provides a rich and nuanced exploration of an under studied topic: the impact of mobility of care (both paid and unpaid) on DDC’s wellbeing.
The discussion part should be strengthened. It is important to have a discussion with the literature in terms of the findings to date versus the research results obtained.
The discussion has also been revised extensively. Far more literature has been added and discussed in relation to the results. Please see pages 12 and 13.
Literature - as already indicated needs to be strengthened. The notation of the bibliography should be refined according to MDPI standards.
We have added a significant number of citations to our manuscript (from 37 to 59). We have also formatted our bibliography according to MDPI formatting (chrome-extension://efaidnbmnnnibpcajpcglclefindmkaj/https://mdpi-res.com/data/mdpi_references_guide_v5.pdf), though if anything is incorrect we would be happy to rectify it.
To summarise.
The topic is interesting, but the research refers to a distant measurement in time (2021), which requires appropriate explanations and justification of the topicality of the topic (summary, methodology).
The background of the research needs a stronger reference to the literature. I recommend developing the introductory section or adding a separate 'literature review' section to more fully explain the research problem and relate it to the literature, strengthening the number of items in the bibliography.
The results of the research should be strengthened on the scientific side - a qualitative study should be performed, statistical tools for the findings should be applied, etc.
The discussion with the literature and the implications should be strengthened.
Literature needs to be strengthened (number of references). Required bibliographic notation according to MDPI standards.
We are so pleased you found this research topic interesting and appreciate your efforts in reviewing our work. We hope that the extensive changes made address all your concerns.
Round 2
Reviewer 3 Report
Comments and Suggestions for Authors
Dear authors, thanks for the additions.
Best wishes, reviewer
Author Response
We are pleased to hear you are satisfied with the revisions made. Thank you for taking the time to review our work. We know the revised manuscript is much improved thanks to your contributions.
The authors
Reviewer 4 Report
Comments and Suggestions for Authors
The authors have improved the article.
Further to the previous review - it is worth considering visualising the main results to improve the readability of the research (qualitative analysis).
The recording of references should be adapted to MDPI standards.
Author Response
Thank you for reviewing our manuscript a second time. We are glad to hear you believe it improved.
We have downloaded the MDPI style format from endnote and re-configured our references based on this style. We have also added a visualization of the results in response to your comment. Pease see p. 7 (or the attached file).
Thank you, again, for all your helpful comments on this manuscript.
